# Propolis and Organosilanes as Innovative Hybrid Modifiers in Wood-Based Polymer Composites

**DOI:** 10.3390/ma14020464

**Published:** 2021-01-19

**Authors:** Majka Odalanowska, Magdalena Woźniak, Izabela Ratajczak, Daria Zielińska, Grzegorz Cofta, Sławomir Borysiak

**Affiliations:** 1Institute of Chemical Technology and Engineering, Poznan University of Technology, Berdychowo 4, 60965 Poznan, Poland; majka.odalanowska@doctorate.put.poznan.pl (M.O.); daria.d.zielinska@doctorate.put.poznan.pl (D.Z.); 2Department of Chemistry, Faculty of Forestry and Wood Technology, Poznan University of Life Sciences, Wojska Polskiego 75, 60625 Poznan, Poland; magdalena.wozniak@up.poznan.pl; 3Department of Wood Chemical Technology, Faculty of Forestry and Wood Technology, Poznan University of Life Sciences, Wojska Polskiego 28, 60637 Poznan, Poland; grzegorz.cofta@up.poznan.pl

**Keywords:** wood/polypropylene composites, wood treatment, propolis, mechanical properties, supermolecular structure

## Abstract

The article presents characteristics of wood/polypropylene composites, where the wood was treated with propolis extract (EEP) and innovative propolis-silane formulations. Special interest in propolis for wood impregnation is due to its antimicrobial properties. One propolis-silane formulation (EEP-TEOS/VTMOS) consisted of EEP, tetraethyl orthosilicate (TEOS), and vinyltrimethoxysilane (VTMOS), while the other (EEP-TEOS/OTEOS) contained EEP, tetraethyl orthosilicate (TEOS), and octyltriethoxysilane (OTEOS). The treated wood fillers were characterized by Fourier transform infrared spectroscopy (FTIR), atomic absorption spectrometry (AAS), and X-ray diffraction (XRD), while the composites were investigated using differential scanning calorimetry (DSC), X-ray diffraction (XRD), and optical microscopy. The wood treated with EEP and propolis-silane formulations showed resistance against moulds, including *Aspergillus niger*, *Chaetomium globosum*, and *Trichoderma viride*. The chemical analyses confirmed presence of silanes and constituents of propolis in wood structure. In addition, treatment of wood with the propolis-silane formulations produced significant changes in nucleating abilities of wood in the polypropylene matrix, which was confirmed by an increase in crystallization temperature and crystal conversion, as well as a decrease in half-time of crystallization parameters compared to the untreated polymer matrix. In all the composites, the formation of a transcrystalline layer was observed, with the greatest rate recorded for the composite with the filler treated with EEP-TEOS/OTEOS. Moreover, impregnation of wood with propolis-silane formulations resulted in a considerable improvement of strength properties in the produced composites. A dependence was found between changes in the polymorphic structures of the polypropylene matrix and strength properties of composite materials. It needs to be stressed that to date literature sources have not reported on treatment of wood fillers using bifunctional modifiers providing a simultaneous effect of compatibility in the polymer-filler system or any protective effect against fungi.

## 1. Introduction

The progressing depletion of fossil fuels as well as increasing environmental awareness and green consumerism result in the development of new products such as, e.g., wood/polymer composites (WPC). For over twenty years, composites of thermoplastic polymers with natural components have been applied in various branches of industry, including the automotive and construction industries [1,2]. Lignocellulose fillers exhibit many advantageous properties such as weight saving, good acoustic and thermal insulation, nonabrasive effect, and good availability [3]. However, these fillers have one important disadvantage, high hydrophilicity, causing poor adhesion to non-polar polymers such as, e.g., polypropylene, as well as sparse dispersion of filler particles in the polymer matrix. For this reason, it is essential to introduce certain modifications in order to improve interface interactions in the composite system. Various methods are used to enhance compatibility between lignocellulose fillers and polymer matrices, e.g., chemical modification [4,5,6], surface grafting of polymers onto fillers [7], introduction of compatibilisers such as maleated polymer [8], and treatment with coupling agents [9,10,11,12]. 

Silanes are effective coupling agents, which are extensively used to enhance polymer–filler interactions [13,14,15,16,17]. These compounds enter condensation reactions with hydroxyl groups of wood, at the same time causing entanglement of polymer matrix chains with xylem fibers. The mechanism results from the fact that hydrolysable groups of silane may hydrolyze, as a result forming silanol, which in turn may react with hydroxyl groups of wood, while organofunctional groups may react with polymer chains [11]. The selection of an organofunctional group is determined by the need to ensure good compatibility with the polymer.

Impregnation of wood with silicon compounds improves its dimensional stability and resistance to weather conditions [18,19,20,21]. Kim et al. (2011) [22] observed that silane treatment significantly improved tensile, flexural, and impact strength of polypropylene composites. Moreover, Ichazo et al. (2001) [23] found an increase in the moduli and tensile strength of wood/polypropylene composites modified with silanes, which was explained by improved dispersion of filler particles in the polymer matrix. Wood treatment using silanes also caused decreased water adsorption, increased thermal stability of the PP matrix, and more homogeneous morphology [3,24]. In turn, Cichosz et al. (2019) reported increased thermal resistance and particle size of cellulose fibers, as well as reduced mass loss as a result of modification with silane coupling agents. 

Another factor causing limitations in the application of WPC composites is related with the relatively low resistance of wood to biological factors, including fungi. Therefore, in recent years, literature sources have reported growing interest in wood impregnation applications of natural products and synthetic compounds characterized by low impact on human health and the environment, including essential oils, chitosan, and terpenes [20,25,26,27,28]. Another example in this respect may be provided by propolis, a natural substance with various biological properties, such as antifungal, antibacterial, antioxidant, and antiviral [29,30,31,32,33]. Compared to untreated wood, the material treated with propolis extracts exhibits activity against decay fungi, such as *Coniophora puteana*, *Trametes versicolor*, and *Neolentinus lepiseus* [32,33]. Literature data indicated that the extract of Polish propolis at a concentration above 12% limited the decay of pine wood caused by *C. puteana* [33]. Propolis extract has also been used as a constituent of wood impregnation preparations. These formulations consisting of propolis extract, silver nanoparticles, and chitosan limited decay of pine wood caused by *T. versicolor* when compared to untreated wood [34,35]. In turn, wood impregnated with two propolis-silane formulations, namely, EEP-VTMOS/TEOS (propolis extract (EEP) with vinyltrimethoxysilane and tetraethyl orthosilicate) and EEP-MPTMOS/TEOS (EEP with 3-(trimethoxysilyl)propyl methacrylate and tetraethyl orthosilicate), limited the activity of *C. puteana* even when the wood samples were subjected to leaching with water [36]. In addition, chemical analyses (AAS, XRF, and NMR) confirmed that constituents of the propolis-silane formulations formed permanent bonds with wood [36]. Moreover, the propolis extract was a component of wood protection formulation containing an ethanolic extract of Polish propolis together with caffeine and silicon compounds (methyltrimethoxysilane and octyltriethoxysilane), which inhibited growth of a brown-rot fungus *C. puteana* [37]. 

The aim of this study was to determine and characterize the supermolecular structure, phase changes, and selected physicochemical properties in composites of polypropylene with pine wood treated using propolis-silane dual component modifiers. To the best of our knowledge, propolis extract and propolis-silane dual modifiers have not been used as wood modifiers applied to composites materials. Therefore, the originality of this study consists in the evaluation of the effect of propolis-silane modifiers of a wood filler with a difunctional action (antifungal activity, compatibility action) on the structure as well as thermal and mechanical properties of wood/polypropylene composites. 

## 2. Materials and Methods

### 2.1. Materials

In this study, Scots pine sapwood (*Pinus sylvestris* L.) was used in the form of sawdust with a grain size of 0.5 mm and wood veneer samples of 20 mm× 20 mm× 0.6 mm. 

A commercially available polypropylene Moplen HP456J produced by Basell Orlen Polyolefins (Płock, Poland) with the melt flow index of 3.4 g/10 min (at 230 °C and 2.16 kg), isotacticity of 95%, and T_m_ = 163–164 °C was used as the polymeric matrix.

Silicon compounds (tetraethyl orthosilicate, vinyltrimethoxysilane, and octyltriethoxysilane) were purchased form Sigma Aldrich (Darmstadt, Germany). The ethanolic extract of Polish propolis was provided by PROP-MAD (Poznań, Poland). Ethanol was purchased from Avantor Performance Materials (Gliwice, Poland). Nitric acid and KBr were purchased from Sigma Aldrich (Darmstadt, Germany).

### 2.2. Wood Treatment

The first formulation used for wood impregnation (TEOS/VTMOS) consisted of 5% tetraethyl orthosilicate (TEOS) and 5% vinyltrimethoxysilane (VTMOS). The other formulation (TEOS/OTEOS) contained 5% tetraethyl orthosilicate (TEOS) and 5% octyltriethoxysilane (OTEOS). The solvent used to prepare silane formulations was 70% ethanol. In the next stage of this study, 70% ethanol was replaced with the ethanolic extract of Polish propolis (EEP) at a 15% concentration. The first propolis-silane formulation consisted of 15% EEP, tetraethyl orthosilicate (TEOS), and vinyltrimethoxysilane (VTMOS) at a 5% concentration (EEP-TEOS/VTMOS), while the other formulation contained 15% EEP, tetraethyl orthosilicate (TEOS), and octyltriethoxysilane (OTEOS) at a 5% concentration (EEP-TEOS/OTEOS). 

The homogenous pine wood material in the form of sawdust was treated with EEP, silanes, and the propolis-silane formulations (1/25 *w/v*). The reaction was run at room temperature with simultaneous stirring, using a magnetic bar stirrer for 2 h. The wood samples were then filtered and dried in air flow at room temperature. 

### 2.3. Characterisation of Treataed Wood

#### 2.3.1. Fourier Transform Infrared Spectroscopy (FTIR)

Wood samples were mixed with KBr at a 1/200 mg ratio. Spectra were registered using a Nicolet iS5 spectrophotometer by Thermo Fisher Scientific (Waltham, MA, USA) with Fourier transform at a range of 500–4000 cm^−1^ and a resolution of 2 cm^−1^, registering 64 scans.

#### 2.3.2. Atomic Absorption Spectrometry (AAS)

Wood samples (0.5 g) were mineralized with nitric acid (8 mL) in the microwave mineralization system (CEM Corporation, Matthews, NC, USA) and after cooling, the solutions were filtered and diluted to 50.0 mL with deionized water. The concentration of silicon in wood samples was determined using flame atomic absorption spectrometry in an AA280FS spectrometer (Agilent Technologies, Santa Clara, CA, USA).

#### 2.3.3. X-ray Powder Diffraction (XRD)

The supermolecular structure of treated wood was analyzed by means of wide angle X-ray scattering (TUR-M62 diffractometer, Carl Zeiss, Jena, Germany). The diffraction pattern was recorded between 5 and 30° (2θ-angle range) in the step of 0.04°/3 s. The wavelength of the Cu Kα radiation source was 1.5418 Å and the spectra were obtained at 30 mA with an accelerating voltage of 40 kV. Peak deconvolution was performed by the method proposed by Hindeleh and Johnson [38], improved and programmed by Rabiej [39]. After separation of X-ray diffraction lines, the degree of crystallinity (X_c_) was determined by comparing the areas under crystalline peaks and the amorphous curve. The changes in the supermolecular structure of wood were analyzed in the function of the chemical modification process.

#### 2.3.4. Biological Activity of Treated Wood

The analyses were carried out on wood veneer samples of 20 mm × 20 mm × 0.6 mm prepared from pine sapwood. The samples were impregnated by soaking in the solutions of modifying formulations. The samples were soaked for 20 min and next conditioned to constant weight at the relative humidity of 65 ± 5% and the temperature of 20 ± 1 °C. The mycological test of treated wood veneer samples was based on the PN EN ISO-846:2019 standard [40]. The tested samples in 5 replicates were evaluated for resistance against *Aspergillus niger* van Tieghem BAM 4 (ATCC 6275), *Chaetomium globosum* Kunze BAM 12 (ATCC 6205), *Penicillium funiculosum* Thom (ATCC 11979), *Paecilomyces variotii* Bainire BAM 19 (ATCC 18502), *Trichoderma virens* (ATCC 9645) and *Ulocladium atrum*. The samples were placed under sterile conditions on the previously prepared and sterilised Petri dishes filled with the agar substrate (potato dextrose agar, Sigma Aldrich, Darmstadt, Germany) and infected using an aqueous solution of spores of the test fungus. The Petri dishes with samples were incubated for 21 days at a temperature of 28 ± 1 °C and relative humidity above 95%. The degree of fungus colonisation on the sample surface was visually observed at 3, 7, 10, 14 and 21 days of the experiment and individually rated using a four-degree scale (Table 1).

### 2.4. Preparation of Composite Materials

The wood-polypropylene composites containing untreated and treated wood were obtained by the extrusion method. The mixture of polypropylene and 30% wood was mixed in a drum blender for 30 min. This mixture was then conveyed to the feed hopper of a single-screw extruder (Fairex, Le Bourget, France). The length-to-diameter ratio L/D of the extruder was 25. During extrusion the temperatures of the four processing zones were adopted as 140, 180, 190 and 195 °C, respectively, at the die temperature of 190 °C. The extrusion speed was 25 to 30 rpm. Extrusion temperature was kept at less than 200 °C to avoid decomposition and degradation of wood. The extrudate was cooled with 20 °C water after exiting the die and then pelletised into granules. Next, the granules were dried in an oven for 24 h at 60 °C.

The samples used to investigate the material structure and to conduct mechanical strength tests were prepared using an Engel injection moulding machine at the mould temperature of 30 °C. After moulding the specimens for structure analyses were immediately sealed in a polyethylene bag and placed in a vacuum desiccator for a minimum of 24 h prior to structural testing.

### 2.5. Characterisation of Composite Materials

#### 2.5.1. Differential Scanning Calorimetry (DSC)

Differential scanning calorimetry was used to characterize thermal properties of composites. The tests were carried out under dynamic conditions using the Netzsch DSC 200 calorimeter (Netzch Group, Selb, Germany) in argon atmosphere. First, the samples were heated up to 220 °C at a rate of 20 °C/min. In order to remove their previous thermal history they were kept at this temperature for 3 min. In the next stage the samples were cooled down at 5 °C/min to 40 °C. The entire cycle was repeated two times and for calculations the data from the second run were used. The crystallization parameters of WPC with unmodified and modified wood fillers, such as crystal conversion (α), half-time of crystallization (t_0.5_) and crystallization temperature (Tc) were determined. 

#### 2.5.2. Morphological Analysis

The nucleation ability of the polypropylene matrix in the presence of wood fillers was assessed applying the hot-stage polarized light microscopy. For this purpose a Labophot-2 microscope (Nikon, Tokyo, Japan) coupled with a Panasonic CCS camera (Panasonic, Kadoma, Japan) and equipped with a Linkam TP93 heating stage (Linkam, Tadworth, UK) was used. Composite film placed on the microscope slide was heated to a temperature of 200 °C at a rate of 40 °C/min. The sample was heated for 3 min to erase the thermal memory of the polymer. The material was cooled to 136 °C at a rate of 20 °C/min while isothermal crystallisation was run. In the course of the experiment the induction time for the crystallisation process and the transcrystalline growth rate were determined.

#### 2.5.3. Structural Investigations (XRD)

The wide-angle X-ray diffraction was used to determine polymorphic changes as well as changes in the supermolecular structure of isotactic polypropylene in composite materials. Diffractograms were analysed applying the method proposed by Hindeleh and Johnson [38], while the content of the polymorphic β-phase (k) was calculated according to the formula proposed by Turner Jones et al. [41].

#### 2.5.4. Mechanical Properties

Tensile strength properties of WPC were determined using a Zwick Z020 universal mechanical testing machine (Zwick/Roell, Ulm, Germany) and evaluated according to the PN EN ISO 527–3: 2019–01 standard [42]. The tests were performed with a load cell capacity of 20 kN at a cross-head speed of 5 mm/min. The basic strength parameters were determined: Young’s modulus (YM), tensile strength (TS) and elongation at break (EB). 

## 3. Results and Discussion

### 3.1. Characterisation of Treated Wood

In the first stage of the research process the chemical and biological characteristics were determined in pine wood samples treated with silanes, propolis extract and the propolis-silane formulations.

#### 3.1.1. Fourier Transform Infrared Spectroscopy

The Fourier transform infrared spectroscopy was the main tool for assessing interaction between silanes and constituents of the propolis-silane formulations with wood. The main changes found in FTIR spectra of wood treated with silanes (D) and the propolis-silane formulations (C) compared to untreated wood (A) and wood treated with EEP (B) are showed in Figure 1 and Figure 2. 

The presented spectra showed changes in the structure of wood after reaction with EEP, silanes and the propolis-silane formulations compared to the untreated material. However, it is worth emphasizing that the changes in FTIR spectra are limited, especially in the case of wood treated with propolis extract and propolis-silane preparations. This is due to the overlapping of the bands of silicon compounds and propolis extract components. Propolis is a very complex substance that contains over 500 different compounds. Many of these compounds react with the components of the wood.

The intramolecular hydrogen bond was observed at around 3400–3300 cm^−1^. It was also related with the –OH association of cellulose and the aromatic system of lignin. Spectra of silane-treated wood comprised bands at 2950–2850 cm^−1^, which were assigned to C–H stretching vibrations of –CH_3_ groups in silanes [43]. In the spectra of wood treated with silanes (D) and propolis-silane formulations (C) there was a band at 710 cm^−1^, which is typical of bonds between silicon and carbon, where the carbon atom comes from the methoxy group of organosilanes. In the spectra of wood treated with silanes and propolis-silane formulations the band was also observed at 1280 cm^−1^, connected with the vibrations of the Si–C and Si–O groups [44]. The appearance of this band indicated the presence of organosilanes in wood structure. The results of FTIR analysis were compatible with the mechanism of the reaction between silanes and wood, discussed in studies by Sebe et al. [43], Tjeerdsma and Militz [45], Tingaut et al. [46] and Ratajczak et al. [47]. The band observed at 1205 cm^−1^ was related to residual unhydrolysed Si–OCH_3_ groups and their small intensity, suggesting that most of the silane molecules in the reaction had been hydrolysed [48]. In addition, apart from those mentioned above in the spectra of wood treated with silanes and the propolis-silane formulations, the band observed at 1730 cm^−1^ turned out to be characteristic of vibrations of the carbonyl group C=O, originating from wood, and it was present only in the spectrum of untreated wood. The disappearance of this band in the spectra of wood treated with silanes and the propolis-silane formulations confirmed the presence of silicon compounds in wood. Moreover, the band at 1637 cm^−1^ was observed in the spectra of wood impregnated with propolis extract (B) and the propolis-silane formulations (C). This band was responsible for vibrations of C=O, C=C and N–H originating from propolis components (mainly flavonoids), which suggests that components of propolis extract also interact with wood [36].

#### 3.1.2. Atomic Absorption Spectrometry

The concentration of silicon (coming from organosilane compounds) in wood impregnated with silanes and the propolis-silane formulations determined by flame atomic absorption spectrometry is presented in Table 2. 

Analysis of Si concentration in wood samples treated with silicon compounds is a useful tool in assessment their presence in wood structure [36,49,50]. The silicon concentrations in treated wood indicated that wood samples impregnated with both silane-propolis formulations contained higher amounts of Si compared to wood treated with silicon compounds without the propolis extract. The highest silicon concentration was detected in wood impregnated with EEP-TEOS/OTEOS. It may be due to the components of the propolis extract that may act as catalysts, causing the formation of reactive silane forms—silanols. Additionally, it is possible to remove low-molecular components of wood with an ethanol solution, which may increase the effectiveness of silane interaction with the wood surface.

#### 3.1.3. X-ray Diffraction (XRD)

The aim of diffraction analyses was to investigate the supermolecular structure of treated wood. Figure 3 presents X-ray patterns of wood and the material impregnated using EEP and propolis-silane formulations.

In Figure 3 the presented diffractometric curves show a variation in the intensity of diffraction maxima, which is caused by the effect of the propolis extract (EEP) and the propolis complex with added silanes on changes in the supermolecular structure of wood. Analyses of the diffraction maxima observed at the diffraction angle 2θ of 15, 17 and 22.7° showed the presence of polymorphic cellulose form I in the lignocellulose material.

Additionally, when comparing the diffactometric curves the degree of crystallinity (X_c_) was determined for the analysed lignocelullose fillers, as presented in Table 3. 

In wood not subjected to the impregnation process the degree of crystallinity was 50%. It was stated that wood treated using various systems is responsible for an increase in crystallinity of the lignocellulose material. In the case of wood treatment with propolis extract the degree of crystallinity increased by approx. 10% (to 56%) compared to untreated wood. Such a result may be explained by the fact that ethanol used as a solvent of propolis was responsible for the removal of low molecular weight compounds and amorphous substances such as resin acids, free fatty acids, pinosylvin and pinosylvin monomethyl ether from wood [51].

Treatment of pine wood with propolis solutions also containing silanes, as well as those containing only silanol modifiers caused a slight increase in the degree of crystallinity of wood compared to the application of EEP solution (within the range of 58–60%). 

#### 3.1.4. Biological Analysis

The results of mycological tests presented in Table 4 indicate that pine veneer samples treated with silanes (TEOS/VTMOS and TEOS/OTEOS) showed no resistance against tested fungal strains. The surface of these samples after 21 days of the fungal test was covered in over 66% by fungal mycelium. Only, wood impregnated with TEOS/VTMOS exhibited moderate resistance against *T. virens*, while wood treated with TEOS/OTEOS showed low resistance against *T. virens* and *A. niger* (less than 33% of the sample surface was covered by fungal mycelium).

Wood treated with the propolis extract was characterized by higher mould resistance compared to the unprotected control samples and wood treated with silicon compounds. Wood protected with the propolis extract showed high activity against all tested fungal strains, except for *Ch. globosum* and *P. variotii*, where wood surface partially covered by mycelium was observed (≤33%). In turn, wood veneers impregnated with the propolis-silane formulations (EEP-TEOS/VTMOS and EEP-TEOS/OTEOS) exhibited high activity against all tested moulds. 

According to literature data, wood impregnated with alkoxysilanes shows no resistance to moulds, such as *Alternaria alternata*, *Cladosporium herbarum*, *A. niger*, *Penicillium decumbenes* and *Penicillium brevicompactum* [50,52,53,54]. In turn, Ghosh et al. [55] reported that impregnation with an amino-silicone macro-emulsion resulted in a certain resistance to growth of moulds (including *A. versicolor*, *Ulocladium strum* and *Aureobasidium pullulans*) on wood surfaces, whereas wood treated with the same concentrations of silicone quat micro-emulsion and alkyl-modified silicone macro-emulsion exhibited comparatively lower resistance. Reinprecht and Grznárik [53] also stated that wood treated with organosilanes without the –NH_2_ group (methyltrimethoxysilane, vinyltrimethoxysilane and propyltrimethoxysilane) was more accessible for mould attacks (*A. niger* and *P. brevicompactum*) than wood impregnated with 3-aminopropyltrimethoxysilane. Therefore, silicon compounds are used in wood protection as components of impregnating preparations limiting leaching of active substances, such as boron compounds or propolis and caffeine from the wood structure [20,36,37]. In turn, literature data showed that propolis extracts exhibited activity against various strains of moulds, including *A. niger*, *A. fumigatus*, *P. notatum*, *P. italicum*, *P. variotti*, *Ch. globosum* and *Schizophyllum commune* [56,57,58]. The extract of Polish propolis was effective against e.g., *A. niger*, *A. versicolor*, *P. pinophilum* and *T.virens* [59,60]. The results of mould resistance of treated wood veneers confirmed that silicon compounds were not effective against tested fungi. The protective effect against moulds was observed in the case of veneers impregnated with the propolis extract and the propolis-silane formulations. The propolis-silane formulations showed higher activity against tested fungi compared to that of the propolis extract. It may be associated with the synergistic action of propolis providing the antifungal activity and silicon compounds reducing the hydrophilicity of treated wood veneers.

### 3.2. Characteristics of Composite Materials

The observed chemical and biological characteristics of wood treated with propolis extract, silanes and the propolis-silane formulations indicated that wood impregnated with propolis extract and the propolis-silane formulations exhibited promising properties when applied as fillers to wood composites. Therefore, the next stage of the research consisted in the characterization of composites with wood treated using propolis extract and two propolis-silane formulations, namely EEP-TEOS/VTMOS and EEP-TEOS/OTEOS.

#### 3.2.1. Differential Scanning Calorimetry of WPC

Composites and the unfilled PP were analysed by differential scanning calorimetry (DSC) to investigate phase changes taking place in polypropylene in the presence of untreated and treated lignocellulose fillers. The obtained thermograms are presented in Figure 4.

It may be observed that the presence of fillers has a considerable effect on the course of polypropylene crystallisation. The curves obtained for the composite systems are shifted towards higher temperature values compared to the unfilled PP. The greatest difference was recorded for composites with wood treated with the propolis-silane formulations, while no significant differences were found for the sample with propolis (PP + EEP). Based on the graphs kinetic parameters of the crystallisation process were determined (Table 5).

Based on the conducted analyses it was observed that the introduction of an untreated wood filler, as well as that treated with propolis-silane formulations causes an increase in crystallisation temperature compared to the unfilled polymer matrix. In the case of untreated wood temperature T_c_ increased by 2 °C compared to that of polypropylene. The greatest increase in crystallisation temperature was observed for the systems comprising wood impregnated with the propolis-silane formulations (119.5–122.5 °C). It needs to be stressed here that the formulation containing tetraethyl orthosilicate (TEOS), octyltriethoxysilane (OTEOS) and the propolis extract is the most effective modifier in terms of the increase in nucleation activity. 

It is generally known that wood is a material used in the process of heterogenic nucleation of polypropylene [61,62]. Also an increase in crystallisation temperature in composites composed of silane-treated wood has been reported in many literature sources. Gironès et al. [3] observed an increase in T_c_ by almost 10 °C in relation to the polymer matrix for all the systems, irrespective of the applied silane compound, which was related with nucleation properties of these compounds. A considerable change in crystallisation temperature from 113 °C to 126 °C compared to unfilled polypropylene was also recorded by Ichazo et al. [23]. They stressed in their study that such a behaviour is highly advantageous, since it indicates that the application of silanes in the process of WPC manufacture makes it possible to markedly reduce the energy consumption and time of their processing.

The next step in the analysis of DSC results recorded for the produced materials consisted in the determination of the degree of phase conversion (Figure 5) and half-time of crystallization (Table 5). 

It may be observed that the degree of conversion for composite systems containing wood treated using silanol systems with propolis at a given time is markedly greater than that for the unfilled matrix or the system with the propolis extract not containing silanes. The obtained dependencies indicate a high nucleation ability of EEP/silane modifiers in the process of polypropylene crystallisation. 

Table 5 presents values of half-time of crystallization determined based on the conversion curves for tested samples. Treatment of the lignocellulose filler was observed to influence recorded results. All the analysed systems were characterised by a lower half-time of crystallization compared to unfilled PP. The greatest reduction of t_0.5_ was reported for composites with the TEOS/OTES silane complex. However, the application of wood treatment using only the propolis extract (PP + EEP) was connected with a considerable decrease in nucleation activity. Composites with wood treated using propolis extract were characterised by high values of half-time of crystallization (2.35 min), slightly lower than values of the unfilled polymer matrix (2.7 min). 

Recorded DSC results showed that the treatment of lignocellulose fillers considerably influences their nucleation properties and causes changes in the course of phase processes. In order to provide greater insight into these problems polarised light microscopy was used in the next stage of the study. 

#### 3.2.2. Polarized Light Microscopy of WPC (PLM)

Figure 6 presents PLM images recorded during the isothermal crystallisation of polypropylene in the presence of wood fillers.

Analyses of the recorded images showed that treatment of lignocellulose fillers is an extremely important factor influencing the course of the PP crystallisation process. It may be observed that a transcrystalline layer (TCL) is formed in all the samples; however, its rate varies greatly. The modifier most effectively increasing the nucleation ability was the complex of propolis extract with tetraethyl orthosilicate (TEOS) and octyltriethoxysilane (OTEOS), which is confirmed by the recorded greatest nucleation density for this system (Figure 6c). Definitely the lowest efficacy in the modification of transcrystalline structures was found for the composite system with wood treated using the propolis solution (Figure 6b). It needs to be stressed here that the transcrystalline layer is formed on the surface of the filler with high nucleation activity, resulting in the crystallite growth perpendicular to the filler surface. The observed phenomenon is highly desirable in the production process, since it leads to markedly improved mechanical properties of materials compared to those, in which perpendicular crystallisation does not occur [62]. Obtained images confirm that the selection of an adequate modifying agent is an extremely important criterion when producing a filler with a high nucleation activity of the matrix. 

Based on the PLM images the rate of TCL growth was determined in the tested systems (Table 6). For composite materials containing untreated wood it was 5.3 μm/min. A higher value was recorded only for the PP + EPP-TEOS/OTEOS systems (6.2 μm/min). The obtained results confirm earlier observations. 

Silane compounds used to modify lignocellulose fillers cause an increase in their nucleation activity, which is manifested in the much greater TCL growth rate and increased spherulite density. A dependence was observed between the rate of heterogeneous nucleation in the polymer matrix and the filler structure. Conducted analyses showed that wood treatment using the propolis extract resulted in a considerable reduction of nucleation activity, which may be explained by the removal of non-structural and low molecular weight compounds during the modification process, with these compounds being capable of serving the role of active nucleants in the crystallisation process. Also in our previous studies [62] it was observed that the removal of impurities, waxes and fats from the surface of lignocellulose fillers (e.g., in the mercerisation process) results in a reduced activity compared to the unmodified systems. It is of interest that the application of the propolis solution in the modification process, but this time with the silanol system, results in a considerable increase in nucleation activity of the wood surface. Such a situation may be the effect of increased compatibility of wood and the polymer matrix thanks to the generated intermolecular interactions.

#### 3.2.3. X-ray Diffraction Analysis of WPC

X-ray analyses were used to determine polymorphic changes as well as changes in the supermolecular structure of composite materials with lignocellulose fillers before and after modification in relation to the pure polypropylene matrix. Figure 7 presents diffraction curves for composites indicating the presence of two polymorphic forms, α and β, both for the polymer matrix, i.e., isotactic polypropylene, and all the tested composite systems.

In the presented diffraction maxima the α-PP form may be observed at angles 2θ of 14°, 17°, 18.5°, 21° and 22°, while the value of 2θ at 16.2°, marked in Figure 7, corresponds to β-PP. Based on analyses of diffractograms of the composite materials and the polypropylene matrix contents of β-PP were calculated, as given in Table 7.

The content of the β-PP form in the isotactic polypropylene matrix is 7%. Shear forces in the course of processing may be responsible for such an amount of β-PP in the unfilled semicrystalline polymer. The composite systems are characterised by markedly higher values of this polymorphic form. The highest β-PP contents were recorded for the PP + EEP-TEOS/OTEOS system (38%), while the lowest level of this form was found for these composites, which filler was treated using the propolis extract. Analyses of composite materials with the untreated lignocellulose filler (PP + wood) showed a slightly higher content of the β-PP form at 28%, compared to the components after filler treatment using propolis extract and the silane complex (PP + EEP-TEOS/VTMOS), in which a 32% content of the β-PP polymorphic form was recorded.

Formation of this polymorphic phase is dependent on the presence of shear forces during processing of semicrystalline polypropylene, in which the addition of a filler causes intensification of stresses between components, which in turn results in the generation of greater values of the above-mentioned shear stresses. Moreover, the formation of β-PP is also influenced by the course of crystallisation in the isotactic polymer matrix. According to literature reports, the process of crystallisation in this semicrystalline polymer is characterised by the polymorphic transition from the metastable β phase to the stable α phase [63,64,65,66]. 

When analyzing the presented results it was observed that the best efficiency of β-PP formation is found in the composite system with wood treated with the propolis extract and tetraethyl orthosilicate (TEOS) and octyltriethoxysilane (OTEOS) system. Moreover, the results are perfectly correlated with the established kinetic parameters provided by DSC and microscopy. Composites, in which the lignocellulose filler was treated with propolis extract together with the tetraethyl orthosilicate and octyltriethoxysilane complex, exhibited the highest nucleation activity and the greatest rate of crystallisation. This is confirmed by the highest values of crystallisation temperatures, the shortest half-time of crystallization and the greatest formation rate of transcrystalline structures. In the case of composite materials, in which a reduced nucleation ability was observed, resulting also in lower crystallisation temperatures, longer half-time of crystallization and inhibition of TCL formation rate, the formation of lower β-PP levels was confirmed. 

#### 3.2.4. Mechanical Properties of WPC

The aim of the conducted mechanical strength tests was to determine the effect of applied novel wood filler modifications on strength characteristics of the produced polymer composites. Table 8 present strength parameter values: strength at break, Young’s moduli, elongation at break and impact strength.

Based on mechanical tests it may be observed that introduction of raw wood filler and wood treated with propolis extract to the polymer matrix causes a slight increase in tensile strength (by approx. 5%). Treatment of wood with a simultaneous application of propolis and silanol systems provides a considerable improvement of strength properties. Composites with such fillers are characterised by an over 20% greater strength at break (PP + EPP-TEOS/OTEOS) and an approx. 15% greater strength at break (PP + EPP-TEOS/VTMOS) compared to the unfilled polypropylene matrix. It also needs to be stressed that for these composite materials a considerable increase was found in their moduli of elasticity (by approx. 50%). In the case of composites with raw wood and those treated solely with EPP the values of Young’s moduli were slightly lower. Very good strength characteristics of composites treated with bifunctional propolis/silane systems result from increased interphase polymer-filler interactions. Very interesting results were obtained for elongation at break and impact strength. Composites with raw wood and wood treated with propolis extract exhibited very low values of strain and relatively low impact strength. Such results are typical for composites of thermoplastic polymers with lignocellulosic fillers described in literature [67,68,69,70], which are characterised by poor adhesion. However, in the case of composites treated by propolis-silane preparations very high values of elongation at break and high impact strength were recorded, which was particularly evident for the PP + EPP-TEOS/OTEOS system. Increased values of elongation at break may be explained by the effect of increased interactions at the interface. However, it needs to be stressed that in the case of literature results reported to date and concerning composites modified solely with silanes such a marked increase in elongation at break has not been observed. This interesting effect recorded in our study may be explained by the potential synergistic effect. The first factor resulting in increased elasticity of composite systems may be connected with the fact that at the application of a bifunctional modifier during modification the small molecular weight compounds are removed from the filler surface by the alcohol propolis extract, which as a consequence promotes the appropriate reaction of silanes with cellulose and generates increased efficiency of wood hydrophobisation. Another factor is related with the supermolecular structure of produced composites. Composites with the filler modified using the bifunctional system (EPP/silanes) are characterised by considerable amounts of the β-PP form, which is particularly evident for the PP + EPP-TEOS/OTEOS system. Literature sources [6,71,72] reported that this polymorphic form of polypropylene exhibits greater elasticity and increased impact strength compared to the α-PP form. Summing up, treatment of wood using the novel modification system generates the β-PP polymorphic form, which considerably influences values of strain and impact strength of the produced composite materials. 

## 4. Conclusions

This paper presents the effect of pine wood treated using novel propolis-silane formulations on properties of the resulting wood/polypropylene composites. In the first stage of the study the chemical and biological characteristics of wood treated with silanes (TEOS/OTEOS and TEOS/VTMOS), the propolis extract and the propolis-silane formulations (EEP-TEOS/OTEOS and EEP-TEOS/VTMOS) were determined. The chemicalanalyses confirmed presence of silanes and constituents of propolis in wood structure. The bands of Si–C and Si–O originating from silicon compounds and the bands associated to propolis components were observed in the spectra of treated wood. Treatment of wood using the propolis extract and propolis-silane formulations caused changes in the structure of wood, including e.g., an increased degree of crystallinity. Moreover, veneer samples impregnated with the propolis extract and the propolis-silane formulations exhibited resistance against moulds, such as *A. niger* and *T. virens* compared to untreated samples or those treated with silanes without the propolis extract.

In the second stage of the study changes in the supermolecular structure as well as thermal and mechanical properties of the composites containing polypropylene and wood treated with the propolis extract and the propolis-silane formulations were determined. Differential scanning calorimetry (DSC) showed that the presence of fillers had a considerable effect on the course of polypropylene crystallisation. The introduction of a wood filler treated with propolis-silane formulations caused an increase in crystallisation temperature and the degree of conversion, which indicates a high nucleation ability of applied modifiers, particularly EEP-TEOS/OTEOS. A high nucleation activity of the filler treated with EEP-TEOS/OTEOS was also confirmed by the results provided by polarised light microscopy (PLM), which showed that the polypropylene composite with wood treated using this preparation exhibited the highest efficacy in the modification of transcrystalline structures, which was manifested in markedly higher values of TCL growth rate compared to the other composite systems. The composite system with the EEP-TEOS/OTEOS treated wood also exhibited the best efficacy of β-PP formation, as well as very good strength properties compared to the other systems. 

Summing up, wood modification using propolis extract and propolis-silane formulations affected the structure, thermal and mechanical properties of wood/polypropylene composites. Obtained fillers with the bifunctional action (antifungal activity and compatibility action) may be added to the polymer matrix in order to prepare green and bio-friendly composites for various applications. 

## Figures and Tables

**Figure 1 materials-14-00464-f001:**
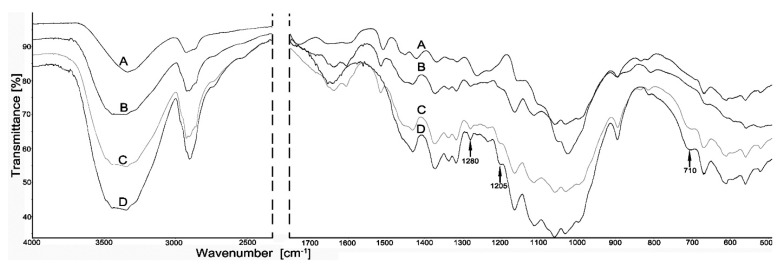
Spectra of wood (A), wood treated with: EEP (B), EEP-TEOS/OTEOS (C) and TEOS/OTEOS (D).

**Figure 2 materials-14-00464-f002:**
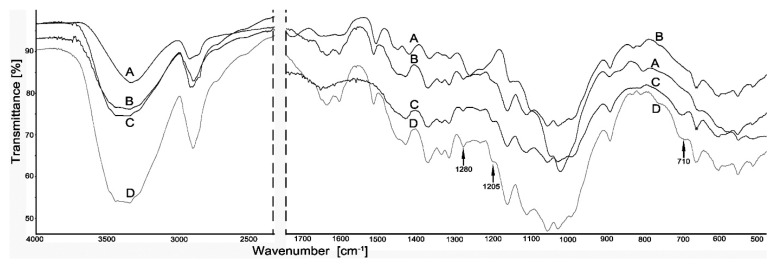
Spectra of wood (A), wood treated with: EEP (B), EEP-TEOS/VTMOS (C) and TEOS/VTMOS (D).

**Figure 3 materials-14-00464-f003:**
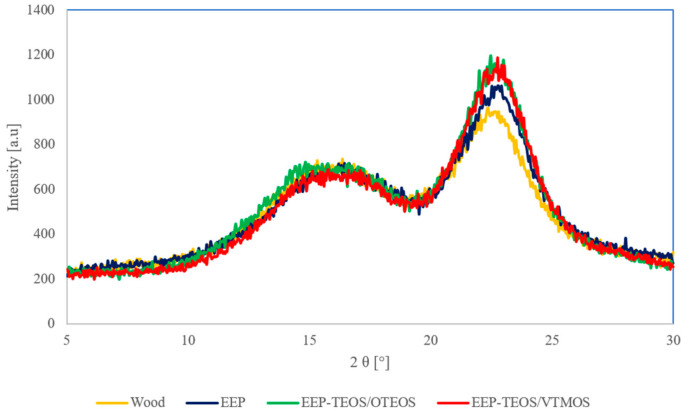
Diffractograms of wood and wood impregnated with EEP, EEP-TEOS/VTMOS and EEP-TEOS/OTEOS.

**Figure 4 materials-14-00464-f004:**
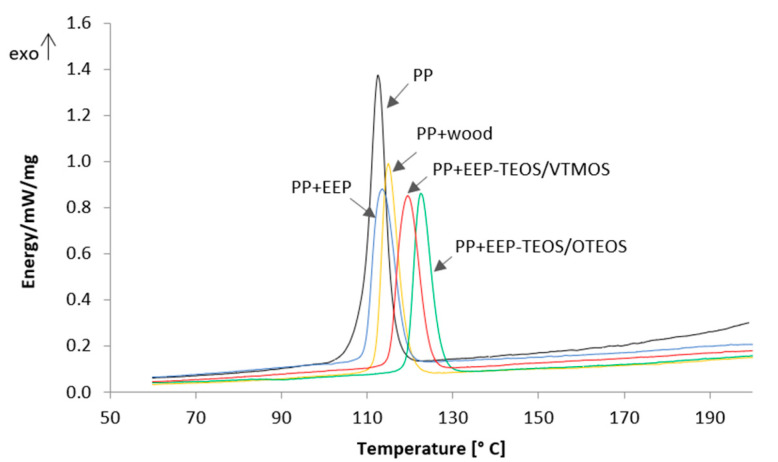
DSC exotherms of WPC.

**Figure 5 materials-14-00464-f005:**
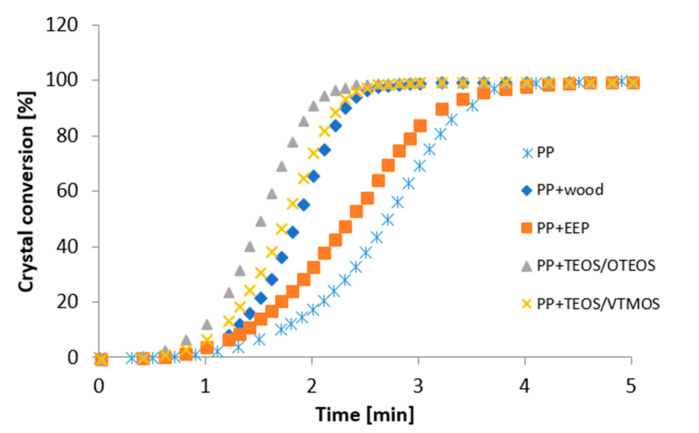
Crystal conversion of PP and WPC.

**Figure 6 materials-14-00464-f006:**
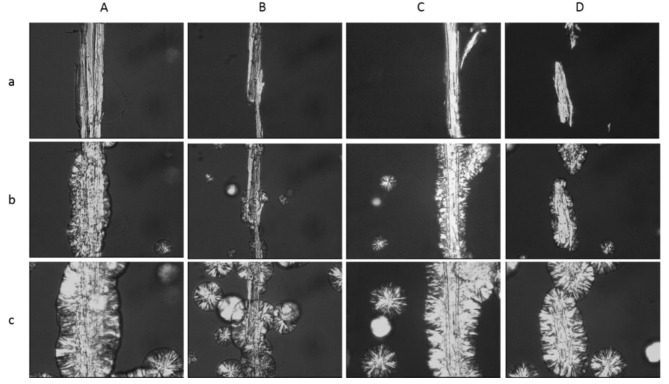
PLM images registered at 136 °C for (**A**) PP + wood, (**B**) PP + EEP, (**C**) PP + EEP-TEOS/OTEOS, (**D**) PP + EEP-TEOS/VTMOS after (**a**) 0 min, (**b**) 3 min, (**c**) 6 min.

**Figure 7 materials-14-00464-f007:**
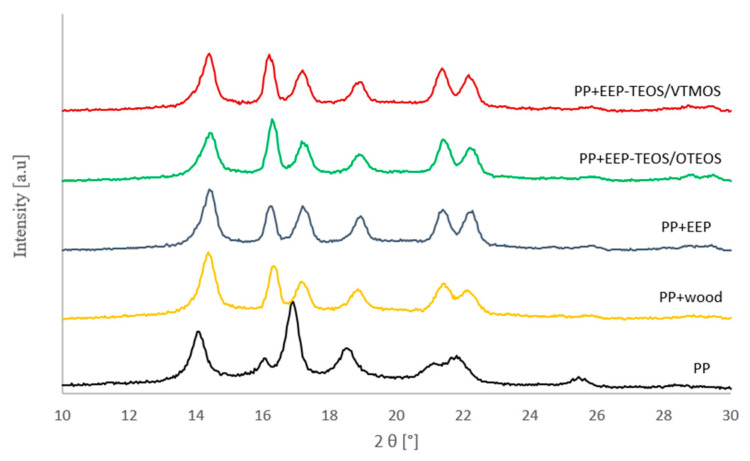
X-ray diffraction pattern of composite materials.

**Table 1 materials-14-00464-t001:** Mould resistance rating scale.

Index	Degree of Sample Colonisation
−1	No sign of mycelium growth on sample, there is a zone of inhibition on the medium between the sample and mycelium
0	No sign of mycelium growth on sample, there is no zone of inhibition on the medium between the sample and mycelium
1	Less than 33% of the sample surface colonised by the tested fungus mycelium
2	More than 66% of the sample surface colonised by the tested fungus mycelium

**Table 2 materials-14-00464-t002:** Silicon concentration in treated wood.

Modified Wood	Silicon Concentration (mg/kg)
TEOS/VTMOS	183.5 ± 4.6
TEOS/OTEOS	186.0 ± 3.8
EEP-TEOS/VTMOS	234.1 ± 7.0
EEP-TEOS/OTEOS	284.4 ± 2.7

**Table 3 materials-14-00464-t003:** Crystallinity grade for lignocellulosic fillers.

Wood Filler	X_c_ (%)
Wood	50
EEP	56
TEOS/OTEOS	58
TEOS/VTMOS	58
EEP-TEOS/OTEOS	60
EEP-TEOS/VTMOS	59

**Table 4 materials-14-00464-t004:** The results of mycological test of treated wood veneers.

Fungal Strain	Modified Wood					
	TEOS/VTMOS	TEOS/OTEOS	EEP	EEP-TEOS/VTMOS	EEP-TEOS/OTEOS	CONTROL
*A. niger*	2	1	0	0	0	2
*Ch. globosum*	2	2	1	0	0	2
*P. funiculosum*	2	2	0	0	0	2
*P. variotii*	2	2	1	0	0	2
*T. virens*	1	1	0	0	0	2
*U. atrum*	2	2	0	0	0	2

**Table 5 materials-14-00464-t005:** Parameters of crystallization of PP and their composites.

Composites	Crystallization Temperature T_c_(°C)	Half-Time of Crystallization t_0.5_(min)
PP	113.0	2.7
PP + wood	115.0	1.85
PP + EEP	113.5	2.35
PP + EEP-TEOS/OTEOS	122.5	1.5
PP + EEP-TEOS/VTMOS	119.5	1.7

**Table 6 materials-14-00464-t006:** TCL growth rate in composite systems.

Composites	Layer Growth Rate of TCL μm/min
PP + wood	5.3
PP + EEP	3.2
PP + EEP-TEOS/OTEOS	6.2
PP + EEP-TEOS/VTMOS	4.3

**Table 7 materials-14-00464-t007:** Contents of the β-PP form in composite materials.

Composites	Content of the β-PP Form (%)
PP	7
PP + wood	28
PP + EEP	23
PP + EEP-TEOS/OTEOS	38
PP + EEP-TEOS/VTMOS	32

**Table 8 materials-14-00464-t008:** Mechanical properties of PP and wood/PP composites.

Composites	Tensile Strength (MPa)	Young Modulus (GPa)	Elongation at Break (%)	Impact Strength (kJ/m^2^)
PP	30.8 ± 0.09	1.24 ± 0.11	1.24 ± 0.11	56.1 ± 0.55
PP + Wood	32.4 ± 0.32	2.24 ± 0.18	2.24 ± 0.18	22.7 ± 0.37
PP + EEP	31.6 ± 0.35	2.19 ± 0.26	2.19 ± 0.26	24.2 ± 0.42
PP + EEP-TEOS/OTEOS	38.2 ± 0.29	2.46 ± 0.19	2.46 ± 0.19	39.6 ± 0.33
PP + EEP-TEOS/VTMOS	35.9 ± 0.43	2.38 ± 0.23	2.38 ± 0.23	31.9 ± 0.39

## Data Availability

Data is contained within the article.

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
