# Peer review of "Propolis and Organosilanes as Innovative Hybrid Modifiers in Wood-Based Polymer Composites"

_materials, 2021, doi:10.3390/ma14020464_

Round 1

Reviewer 1 Report

It is just another article where thermoplastic polymer is filler with reinforcement. However, the small variations the authors made could attract other researchers to read.

Figure 1 and figure 2: quality poor

Changes in FTIR spectra are limited, why?

Why is there a huge difference in silicaon concentration between different modified wood?

Author Response

Dear Reviewer,

We are very grateful for the in-depth review and the piece of advice which was a valuable contribution to our study. Thank you very much for your remarks.

Below are our answers to your questions and remarks concerning our manuscript entitled:

“Propolis and Organosilanes as Innovative Hybrid Modifiers in Wood-based Polymer Composites”.

Manuscript ID: materials-1054955

Our comments and changes are noted below, and are marked in yellow and blue in the manuscript.

Responses of the reviewer’s comments

Reviewer 1

We would like to thank you for your helpful and constructive comments and suggestions. We very much appreciate your suggestions, which have been very helpful in improving the manuscript. Below you will find our answers and replies to your points.

Remark 1:

Figure 1 and figure 2: quality poor.

Answer:

According to your suggestion we have added these figures in higher quality (files JPG and PDF).

Remark 2:

Changes in FTIR spectra are limited, why?

Answer:

The changes in FTIR spectra are limited, especially in the case of wood treated with propolis extract and propolis-silane preparations. This is due to the overlapping of the bands of silicon compounds and propolis extract components. Propolis is a very complex substance that contains over 500 different compounds. Many of these compounds (which are present in very low concentrations in EEP) react with the components of the wood. However, the most important bands, which are discussed in the manuscript, are marked with arrows and described with wavenumber in Figures 1 and 2. In the case of marked bands, changes in intensity of bands are observed in the spectra of wood treated with the applied formulations compared to the wood spectra. However, according to Referee’s suggestion, we included new sentences into the revised manuscript (blue color).

Remark 3:

Why is there a huge difference in silicaon concentration between different modified wood?

Answer:

The differences in the concentration of silicon in the impregnated wood may be related to the reactivity of the mixture of silicon compounds used for impregnation and the addition or absence of propolis extract. Many literature data show that various silicon compounds show variable reactivity with wood (Van Opdenbosch, D.; Dorstein, J.; Klaithong, S.; Kornprobst, T.; Plank, J.; Hietala, S.; Zollfrank, C.Chemistry and water-repelling properties of phenyl-incorporating wood composites. Holzforschung 2013, 67, 931–940; Aaserud, J.; Larnøy, E.; Glomm, W. Alternative systems for wood preservation, based on treatment with silanes. Balt. Netw. Wood 2009, 21–26; Baur, S.I.; Easteal, A.J. Improved photoprotection of wood by chemical modification with silanes: NMR and ESR studies. Polym. Adv. Technol. 2013, 24, 97–103; Woźniak, M.; KwaÅ›niewska-Sip, P.; Krueger, M.; Roszyk, E.; Ratajczak, I. Chemical, biological and mechanical characterization of wood treated with propolis extract and silicon compounds. Forests 2020, 11). Moreover, in the wood treated with silicon compounds and propolis extracts, a higher concentration of Si was found than in wood treated only with silicon compounds. This may be due to the components of the propolis extract that may act as catalysts, causing the formation of reactive silane forms - silanols.

Additionally, in the corrected manuscript we included some more precise information about difference in silicon concentration (blue color).

We hope that our corrections could be accepted by you and manuscript will be transferred to publisher. Again, thank you very much for your deep analysis of our work.

Yours sincerely,

Izabela Ratajczak (on behalf of all authors)

Reviewer 2 Report

The manuscript entitled “Propolis and Organosilanes as Innovative Hybrid Modifiers in Wood-based Polymer Composites” presents the outcome of treating wood with bio-fiendly materials based on propolis and silane, i.e. the increased resistance against fungi, with no cost on the mechanical properties of the composites. This work adds to the effort of obtaining materials having hydrophobic properties and improved durability under weather conditions. The stucture of the composites is investigated by standard physico-chemical methods, the infra-red and atomic spectroscopy, whereas X-Ray diffraction and PLM prove crystallite formation across the fibers.The paper is well written and the results are clearly presented and discussed, therefore I recommend this work to be accepted for publication.

Author Response

Dear Reviewer,

We are very grateful for the in-depth review and the piece of advice which was a valuable contribution to our study. Thank you very much for your remarks.

Below are our answers to your questions and remarks concerning our manuscript entitled:

“Propolis and Organosilanes as Innovative Hybrid Modifiers in Wood-based Polymer Composites”.

Manuscript ID: materials-1054955

Our comments and changes are noted below, and are marked in yellow and blue in the manuscript.

Reviewer 2

We would like to thank the Reviewer for taking the time and effort necessary to review the manuscript. We sincerely appreciate your positive and very kind opinion on our article.

Yours sincerely,

Izabela Ratajczak (on behalf of all authors)

Reviewer 3 Report

The novelty of the present paper is the use of Propolis as modifier for wood fillers in WPC. The results are so clearly positive and consistent in many aspects as I have seen very seldom in research work on materials. I am confident and I trust that the researchers have worked carefully according to scientific principles and that these results are not misleading.

The manuscript is of very good quality, written in excellent English. Results are clearly described in tables and figures. A few editorial remarks remain before the paper can be published:

L107 correct plural ….wood veneer samples….

L193: correct instrument description ….Labophot-2 microscope (Nikon…)

L291: delete ‘The’ in the subheading 3.1.4

L296: correct ….showed low resistance…

L326: correct …reducing the hydrophilicity of treated wood veneers.

L328: correct plural …Characteristics of Composite Materials

L333 correct …with wood treated using….

Fig. 4: replace DSC in the y-axis title to the measured property (energy), improve line markers to enable reading the plot in black and white

L345 I think it is not correct to write here of a complex of propolis with silanes because it may be a blend and not a real complex structure

Table 5: Explain Tc and t0.5 in table caption or a legend, correct the comma in t0.5 to English spelling (.)

L369, 378, 380, 384: find a correct term for ‘half-time crystallisation time’ and use the same one throughout the manuscript

Fig. 5: delete ‘The’ in the figure caption

L405: what is meant with ‘active filler’? Perhaps you mean the modified filler/modified wood particles?

L406: correct ‘processing process’ to ‘production process’

L489: add literature citations for this statement

L504: add literature citations for this statement

Blanks are missing in many places, such as: L84, 209, 213, 217, 231, 234, 238, 242 (2x), 260, 337, 408, 410, 483

There is one blank too much in L210 in the unit mm/min

Author Response

Dear Reviewer,

We are very grateful for the in-depth review and the piece of advice which was a valuable contribution to our study. Thank you very much for your remarks.

Below are our answers to your questions and remarks concerning our manuscript entitled:

“Propolis and Organosilanes as Innovative Hybrid Modifiers in Wood-based Polymer Composites”.

Manuscript ID: materials-1054955

Our comments and changes are noted below, and are marked in yellow and blue in the manuscript.

Responses of the reviewer’s comments

Reviewer 3

We would like to thank you for your helpful and constructive comments and suggestions. We very much appreciate your suggestions, which have been very helpful in improving the manuscript. Below you will find our answers and replies to your points.

Remark 1:

Fig. 4: replace DSC in the y-axis title to the measured property (energy), improve line markers to enable reading the plot in black and white

and

Table 5: Explain Tc and t0.5 in table caption or a legend, correct the comma in t0.5 to English spelling (.)

Answer:

We are much grateful for this remark. According to Referee’s suggestion in the revised manuscript we corrected quality of the Figures (additionally, Figure 7) and Table 5.

Remark 2:

L345 I think it is not correct to write here of a complex of propolis with silanes because it may be a blend and not a real complex structure.

Answer:

We agree with that valuable remark. You are right. In the corrected manuscript we included some more precise information about modifiers.

The new sentence is as follows:

The greatest difference was recorded for composites with wood treated with the propolis-silane formulations, while no significant differences were found for the sample with propolis (PP+EEP).

Remark 3:

L405: what is meant with ‘active filler’? Perhaps you mean the modified filler/modified wood particles?

Answer:

We would like to apologize for the unclear expression. Indeed, this phrase may raise some doubts. As suggested by the Reviewer, we have changed this sentence of the publication.

New sentence is as follows:

It needs to be stressed here that the transcrystalline layer is formed on the surface of the filler with high nucleation activity, resulting in the crystallite growth perpendicular to the filler surface.

Remark 4:

L489: add literature citations for this statement

L504: add literature citations for this statement

Answer:

We are grateful for this important remark. According to Referee’s suggestion, we included new references into the revised manuscript.

The new references are as follows:

  1. Joseph, P. V.; Mathew, G.; Joseph, K.; Groeninckx, G.; Thomas, S. Dynamic mechanical properties of short sisal fibre reinforced polypropylene composites. Compos. Part A: Appl. Sci. Manuf.2003, 34, 275–290, doi:10.1016/s1359-835x(02)00020-9.
  2. Nygård, P.; Tanem, B.S; Karlsen, T.; Brachet, P.; Leinsvang, B. Extrusion-based wood fibre-PP composites: Wood powder and pelletized wood fibres - a comparative study. Compos. Sci. Technol.2008, 68, 3418-3424, doi:10.1016/j.compscitech.2008.09.029.
  3. Leluk, K.; FrÄ…ckowiak, S.; Ludwiczak, J., Rydzkowski, T.; Thakur, V.K.. The Impact of Filler Geometry on Polylactic Acid-Based Sustainable Polymer Composites. Molecules.2021, 26, 149-167, doi:10.3390/molecules26010149.
  4. Michalska-Pożoga, I.; Rydzkowski, T. The effect extrusion conditions for a screw-disc plasticizing system on the mechanical properties of wood-polymer composites (WPC). Polimery.2016, 3, 202-210, doi:10.14314/polimery.2016.202
  5. Tordjeman, Ph.; Robert, C.; Marin, G.; Gerard, P. The effect of α, β crystalline structure on the mechanical properties of polypropylene. Eur. Phys. J.2001, 4, 459–465, doi:10.1007/s101890170101.
  6. Varga, J. β-modification of isotactic polypropylene: preparation, structure, processing, properties, and application. J. Macromol. Sci. Part B. Phys.2002, 41, 1121-1171, doi:10.1081/MB-120013089.

Remark 5:

A few editorial remarks remain before the paper can be published.

Answer:

We are grateful for all important editorial remarks and we are very sorry for the various mistakes. Of course, all comments were introduced into the revised version of our paper.

We hope that our corrections could be accepted by you and manuscript will be transferred to publisher. Again, thank you very much for your deep analysis of our work.

Yours sincerely,

Izabela Ratajczak (on behalf of all authors)